# 'Long journey to shelter': a study protocol: a prospective longitudinal analysis of mental health and its determinants, exposure to violence and subjective experiences of the migration process among adolescent and young adult migrants in Sweden

Erica Mattelin ,[1] Amal R Khanolkar,[2,3] Frida Fröberg,[1] Linda Jonsson,[4] Laura Korhonen [1,5]

For numbered affiliations see end of article.

**Correspondence to**
Professor Laura Korhonen;
laura.korhonen@liu.se

## ABSTRACT

**Introduction** According to the UN Refugee Agency (UNHCR), around 40% of the 79.5 million forcibly displaced persons in the end of the year 2019 were children. Exposure to violence and mental health problems such as posttraumatic stress disorder are frequently reported among migrant children, but there is a knowledge gap in our understanding of the complex longitudinal interplay between individual, social and societal risk and resilience factors that impact mental health and well-being, quality of life and ability to function and adapt. There is also an urgent societal need to facilitate interdisciplinary and intersectoral collaborative efforts to develop effective methods to prevent, detect and respond to the needs of the migrants. This project will study adolescent and young adult migrants in Sweden using multiple methods such as quantitative analysis of data from a prospective cohort study and qualitative analysis of data gathered from teller-focused interviews. The aim is to understand how different factors impact mental health and integration into the Swedish society. Furthermore, individual experiences related to the migration process and exposure to violence will be studied in detail.

**Methods and analysis** Study participants will include 490 migrants aged 12–25 years recruited through social services, healthcare, social media and the civil society. A subsample of adolescents (n=160) will be re-interviewed after 1 year. Data are collected using structured and semi-structured interviews along with saliva and hair sampling. Measures include sociodemographic data, longitudinal data on mental health and its determinants, including genotypes and stress-hormone levels, access to healthcare and the process of migration, including settlement in Sweden.

**Ethics and dissemination** The Regional Ethics Board of Linköping (2018/292-31 and 2018/504-32) and the National Ethics Board (2019–05473,2020–00949 and 2021-03001) have approved the study. Results will be made available to participants, their caregivers,

## Strengths and limitations of this study

► This is the first prospective longitudinal study in Sweden on newly arrived refugee and asylum seeking adolescents and young adults, and aims to investigate exposure to violence and mental and physical health using face-to-face interviews. However, this protocol will not address cultural, religious, economic or political factors due to limits of possible data collection in interpreter-assisted interviews.

► Using a multimethod approach, including analysis of genotypes and biomarkers, the study will provide a comprehensive set of data that allows a nuanced analysis of within immigrant population as well as context-dependent and time-dependent factors that explain the different outcomes.

► Results can potentially improve care of migrants and aid in developing current intersectoral procedures, including public health policies.

► Overrepresentation of study participants from particular countries could lead to selection bias.

► Loss to follow-up could impact the longitudinal analysis.

professionals working with migrants, researchers and the funders.

## INTRODUCTION

Today millions of children are fleeing their countries, either voluntarily or being forced to. The group of migrant children is heterogeneous and includes asylum seekers and refugees fleeing with their families as well as unaccompanied minors.[1] Most refugees/asylum seekers are forced to flee their homes due to acute threats such as conflicts, famine

and instability, but the migration problem also reflects massive socioeconomic inequalities that exist between different countries and regions.[2 3] Faced with this reality, migration is considered to be one of the greatest complex and current challenges that cannot be tackled quickly by simple measures.[4]

In 2019, of the 79.5 million people forced to flee their homes, 40% were minors.[2] During the peak immigration year of 2015, 70 384 children applied for asylum in Sweden, about half of them were unaccompanied.[5] Thereafter, the number of migrant children in Sweden has gradually decreased reflecting the European Union policy to limit migration.[5 6] In 2020, approximately 13 000 asylum seekers, including 3566 children, came to Sweden. Most of the asylum seeking children originated from Syria, Afghanistan, Iraq and Eritrea.[7]

The definition of international migration encompasses the process of leaving the home country and settlement and integration into the society of destination. In practice, the process is most often intervened with complicated, stressful and potentially hazardous transient phases such as travelling in the hand of illegal smugglers, staying in multinational refugee camps or asylum accommodations and multiple replacements between different countries, accommodations and administrative systems with subsequent interruptions in established human relationships. Thus, migration is associated with several considerable risks such as being targeted by illegal smugglers and other types of crimes,[8] exposure to potentially traumatising events as well as mental and physical health problems,[9–12] including multimorbidity[13] and premature death.[14]

The arrival of migrants may be challenging for the host country. Barajas-Gonzalez et al[15] showed that immigration enforcement in the USA could lead to severe effects on children's mental health, such as children living with a constant fear of their parents being taken into custody, leading the authors to suggest that the immigration enforcement may become a form of psychological violence. But in general, it is largely unexplored what damage is caused to children by slow and insecure bureaucracy.

Current state of the art has shown that migrant children have experienced potentially traumatising events more often than non-migrant children in high-income countries. Studies also report a higher prevalence of mental health problems, such as posttraumatic stress disorder (PTSD), anxiety and depression, among migrants compared with non-migrant children.[9 16 17] Several studies have demonstrated sex-based or gender-based differences such as high rates of sexual violence against females.[18–20] However, the link between potentially traumatic events and mental health among migrant children has not been fully explained. This is largely because most studies until have been cross-sectional in design, with small sample sizes, and have focused on children that have already been in the host country for a considerable time.

Current research literature has largely focused on risk factors and mental illness.[21] Less attention has been

paid to individual and social resilience factors and their interplay with risk factors, which is important for, among other things, a favourable adaptation to a new environment.[21 22] Resilience entails the achievement of desirable social and emotional adaptation, although the individual was exposed to a great risk.[23] Knowing that exposure to repeated violence is not only a direct danger to a child but also may have long-term consequences for the mental and physical health,[24 25] efforts should be taken to strengthen the child's and family's resilience.

## Objectives

The overall research aim is to study and understand how different factors impact health, well-being and integration of migrants in Sweden. We hypothesise that the refugee and asylum seeking group of children is heterogeneous, including well-adjusted individuals with good health and those with severe problems, multimorbidity and functional impairments. We further hypothesise that exposure to violence, lack of social support and network negatively impact mental health, functional ability and well-being.

This study aims:
► To identify individual-related factors that impact health and well-being. It includes studies on traumatisation, mental health problems, resilience as well as genotypes, stress hormones and other biomarkers from collected biomaterial.
► To study which social factors impact health and well-being. It includes analysis of network and support.
► To describe children's experiences related to the entire migration process.

In general, expected results from this study include estimates of the prevalence of experiences of different types of violence, polyvictimisation and mental health problems, as well as their impact on functional ability and well-being. Also, the roles played by social support as well as different biological factors in this context are elucidated. Qualitative studies are expected to give us information about how adolescents migrants conceptualise their own stories. Mechanisms of, among others, polyvictimisation and resilience and their effects on health will be covered in future studies.

## METHODS AND ANALYSIS
### Design and participants

The term 'migrant' ('barn och unga på flykt' in Swedish) refers to refugees (ie, those defined according to the 1951 Refugee Convention), asylum seekers, family reunification migrants (ie, family to a person with a permanent or temporary residence permit), quota refugees (ie, a person who has been selected by the UNHCR to be resettled to a third country) and undocumented migrants (ie, former asylum seekers who lack proper authorisation to stay). The rationale for including all categories of refugees is to obtain a holistic picture of the current situation as well as to be able to follow similarities and differences

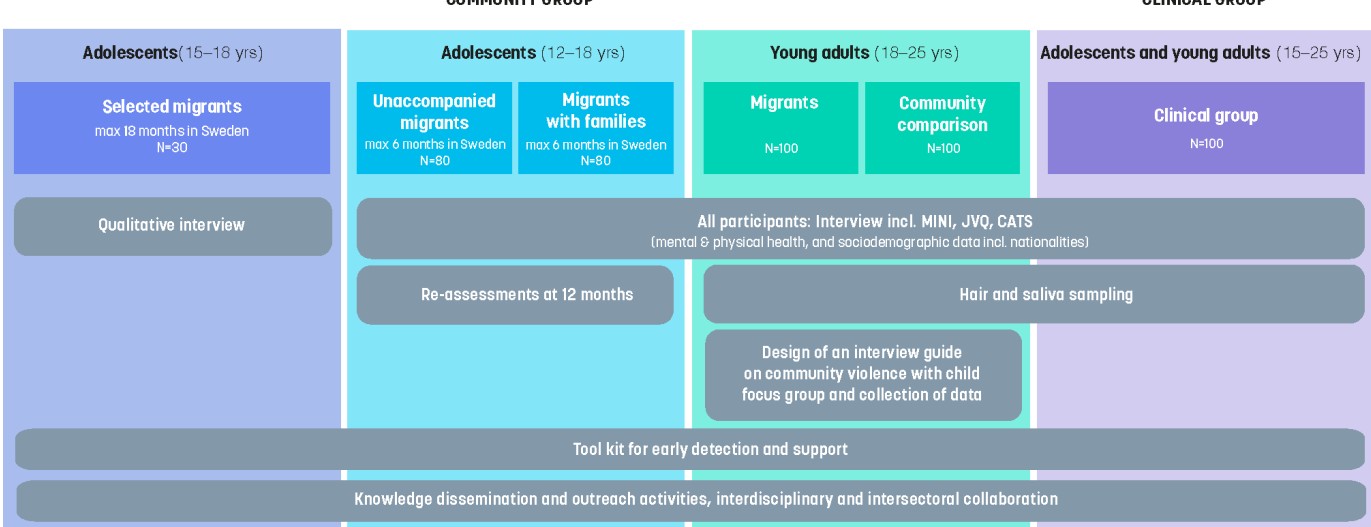

**Figure 1** Summary of the study protocol. CATS, Child and Adolescent Trauma Screen; JVQ, Juvenile Victimisation Questionnaire; MINI, Mini International Neuropsychiatric Interview.

between groups. The population recruited to the study is migrant adolescents and young adults residing in Sweden during 2018–2021. As this is a heterogeneous group, four different study groups will be included involving participants recruited from the community as well as from clinical settings (figure 1).

1. Adolescents (quantitative study): 80 unaccompanied and 80 accompanied adolescent migrants that will be recruited in 2018–2021. Inclusion criteria are migrant, 12–18 years of age, and should not have applied for asylum in Sweden more than 6 months before the study interview. Follow-up interviews are done 1 year after the first interview.
2. Adolescents (qualitative interviews): 30 adolescents for whom the inclusion criteria are 15–18 years of age, are migrant and have applied for asylum in Sweden at a maximum of 18 months before the interview date.
3. Young adult migrants and Swedish-born peers (quantitative study): this includes 100 young adults who migrated to Sweden after 2015. Inclusion criteria are migrants between the ages of 18 years and 25 years living in Sweden. A comparison group consists of a sample of 100 young adults born in Sweden.
4. Clinical sample (quantitative study): this will include 100 adolescent and young adults to be recruited from a trauma unit at a Refugee Medical Centre in Norrköping, Sweden. Inclusion criteria are 15–25 years of age, migrant and living in Sweden.

National statistics on asylum seekers in Sweden are continuously followed to monitor the representativity of the all the subsamples.[26] If needed, targeted recruitments will be initiated.

## Sample size

The aim is to collect data on 490 children and young adults. This sample size allows proposed analyses of primary outcomes with statistical strength of 0.80 and an

α of <0.05 and is based on previous studies with similar methodology,[27–29] which included 95–138 adolescent migrants, with 47–77 adolescent migrants at follow-up assessments. We aim to have a larger sample to be able to do more detailed analyses.

## Recruitment

Participants are recruited across Sweden since refugees/asylum seekers are given a random municipality placement while waiting for a decision on their application for asylum. Recruitment is done via healthcare, social services, schools, social media and the civil society such as non-governmental organisations working with the migrants. The clinical sample is recruited from the Refugee Medical Centre/Region Östergötland and the community comparison group mainly through high schools.

Recruitment of study participants is mainly via people working directly with refugee and asylum seekers who are given leaflets describing the study. Those who agree to participate will be subsequently contacted by the researchers to first obtain informed consent as well as to schedule an interview. All information is given in oral and written form in the participant's native language.

Each interview is scheduled to last for 2 hours and a certified translator is used, if needed. Interviews are held at a location chosen by the interviewee and/or the family.

## Interviews

Semi-structured, translator-assisted interviews are conducted to collect sociodemographic data on country of birth and self-identified ethnicity, family situation, schooling as well as social networks and social support (table 1). Symptoms of mental disorders are assessed using the diagnostic interview MINI-KID/MINI (the Mini International Neuropsychiatric Interview).[30] The presence of PTSD is evaluated using the Child and Adolescent

**Table 1** Data collection and used instruments

| Themes | Variable/measure | Method | Study group(s) |
|---|---|---|---|
| Sociodemographics | Age, sex, country of birth, self-identified ethnicity, etc. | Semi-structured interview based on an interview guide | All except for those participating qualitative interviews |
| Socioeconomic status | Parents' education and occupation | Semi-structured interview based on an interview guide | See above |
| Traumatic events | Potentially traumatic events | JVQ | See above |
| Mental health | Psychiatric symptoms | MINI/MINI-KID | See above |
| | PTSD | CATS and PCL-5 | See above |
| Resilience | Resilience in domains self, family, peers, school and society | ARQ | See above |
| Impairment | Observer rated functional capacity | GAF | See above |
| | Self-rating on impaired functioning | WSAS | See above |
| Well-being | Self-rating on physical health and well-being | WHO-5 Well-Being Index | See above |
| Social network | Family members and contact with friends and family | Question regarding if the child lives with their family, the families' whereabouts and if and how they keep in contact | See above |
| Societal support | Access to care and services | Semi-structured interview based on an interview guide | See above |
| Biological data | Genotype and biomarkers | Hair and saliva sampling and analysis | Young adult cohort and age-matched and sex-matched controls |

ARQ, Adolescent Resilience Questionnaire; CATS, Child and Adolescent Trauma Screen; GAF, Global Assessment of Functionality; JVQ, Juvenile Victimisation Questionnaire; MINI/MINI-KID, Mini International Neuropsychiatric Interview; PCL-5, PTSD Checklist for DSM-5; PTSD, posttraumatic stress disorder; WSAS, Work and Social Adjustment Scale.

Trauma Screen[31] for those under 18 years of age and the PTSD Checklist for DSM-5 for those over 18 years of age.[32] The Juvenile Victimisation Questionnaire is used to measure the number of potentially traumatising events.[33] The questionnaire has been adapted to this study to specifically assess the timing by including the question on whether potentially traumatising events happened before, during or after the process of fleeing one's country of origin. Also, additional questions on issues related to natural disasters, accidents, medical procedures, death of close relatives, poverty, separation from their parents, being captured or imprisoned and human trafficking are included. Resilience is evaluated using the Adolescent Resilience Questionnaire,[34] including five different domains: self, family, peers, school and society. Participant will complete a self-rating on physical health and well-being using the WHO-5 Well-Being Index.[35] The participants' functional capacity is mapped with the Global Assessment of Functionality and Work and Social Adjustment Scale.[36 37] The interviews are verified by an experienced clinical psychologist to ensure good quality. Also, a random selection of 10% of the interviews are quality assessed independently by two members in the research group.

The qualitative interviews are conducted using the 'teller-focused interview' method, which is particularly suitable for studies on sensitive topics and ongoing processes.[38] The interviews are semi-structured and include few overarching questions that covers areas such as the adolescents' experiences in the home country (eg, school, friends, hobbies and well-being), issues related to the entire migration process (eg, reasons to flee, memories from the journey and exposure to violence if any), life during the first 18 months in Sweden and their future aspirations for the 10 years ahead. Total time per assessment is approximately 60 min.

### Biological material
A 3-centimetre long hair sample is collected and stored at room temperature until analysis. Saliva samples are collected using Salivette Cortisol (Sarstedt AG & Co KG, Nurnbrecht, Germany) and stored at −80° until analysis.

### Data analysis plan
Figure 2 provides an overview of the main exposures, outcomes, confounders and analysis that will be conducted. Descriptive analysis of collected data will include presenting continuous variables as mean values

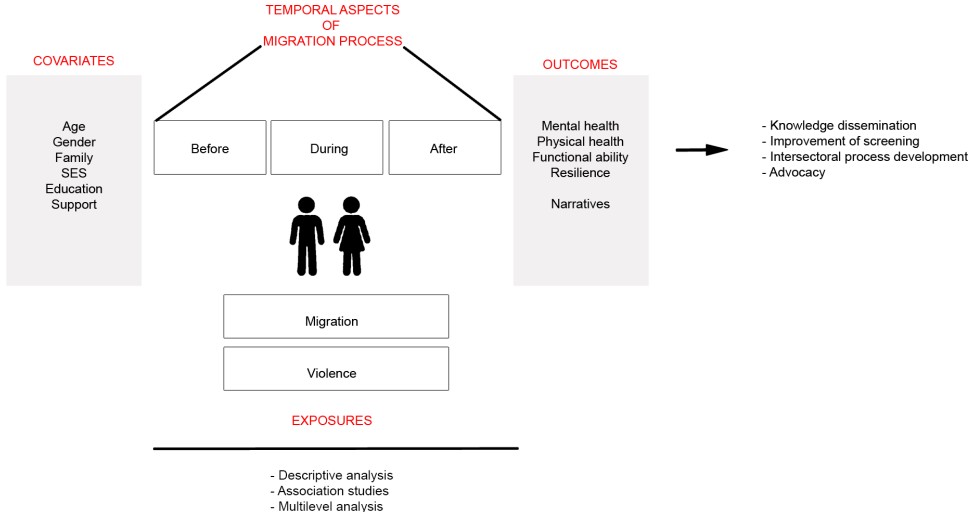

**Figure 2** Summary of the main exposures, outcomes, confounders and analysis as well as the expected impact of the study.

with SD and categorical variables as proportions for the entire study sample but also by important covariates like sex, unaccompanied/accompanied and country of origin. Associations of interest between various variables will be analysed using univariate linear regression or $\chi^2$ tests for differences of proportions for continuous and categorical variables, respectively.

A series of regression models will be performed to study associations between exposures (eg, exposure to violence and asylum status) and outcomes of interest (eg, mental health, functional ability and comorbidity (eg, co-occurrence of mental and physical health conditions)) using multivariate linear and logistic regression for continuous and categorical outcomes, respectively. Each set of regression models will be adjusted for potential confounders such as sex, family socioeconomic position, age and country of origin.

An analysis of individual, environmental and social factors that explain better adaptation, resilience and level of function will be conducted. First, a descriptive analysis of the sociodemographic background, exposure to violence, the prevalence of mental and physical health problems, well-being and resilience factors, in the adolescent cohort and young adult sample will be performed. Second, the prevalence of violence, and mental health problems, and the association between violence and mental health problems in the different phases of the migration process in both groups will be estimated. Besides, the potential moderating effect of resilience factors on the association between exposure to violence and mental health problems in the adolescent cohort will be analysed. Third, because it is assumed that the migration and resilience processes, and the development of mental health problems, are complex and potentially non-linear, multidirectional and multilevel, we will also conduct a multilevel analysis of longitudinal data.

Qualitative interviews will be analysed according to the principles for thematic analysis.[39] Genotypes and biomarkers such as stress hormone levels and immuno-profiles will be analysed using standard methods.

## Gender aspects

We know, based on previous research, that the associations between certain risk factors, such as traumatic events, and mental health differs between the sexes.[40] Such differences are often attributed to genetic or biological factors, and sociological processes, such as conforming to gender normative behaviour or how girls and boys are educated differently, for example. When it comes to the experiences and health of migrant children, less effort has been made to understand the importance of gender. In our study, we will initially stratify analyses by sex and potential sex differences will then be interpreted with a gender perspective, with the intention of understanding and explaining the importance of gender in the link between exposure and outcome. We will also take into account that our respondents may choose to identify themselves by ethnicity, sexuality and family socioeconomic background.

## Methodological issues

The longitudinal study design will allow analysis of changes over time in the same study participant, providing stronger evidence for causality than could be obtained from a cross-sectional design. Combined interview and biological data will provide a versatile set of information that can be used to address different kinds of research questions. Use of standardised structured and semi-structured interviews and validated instruments will also allow comparisons to other studies.

The primary limitation of this study may arise if migrants from a few countries with a high certainty of getting asylum such as Syria will be overrepresented in the study. Furthermore, we could also have the risk of concerned parents or social workers not letting all the adolescents participate because of fear of causing distress. This may

exclude certain groups such as those with existing mental health problems and/or exposed to violence.

Yet another potential limitation is the loss to follow-up because migrants might leave the country. Efforts to minimise loss to follow-up will include respecting the time commitment of patients, formal consented tracking procedures of patients enrolled, multiple contacts for arranging follow-up and flexible hours and places for interviews.

## Data management, storage and security
Throughout the course of this project, research data are collected and processed according to FAIR (findability, accessibility, interoperability and reusability) data principles. A detailed logbook ensures to minimise the risks of data errors and inaccuracies as well as data sharing and reuse. Data set is password protected, pseudonymised and coded for storage and analysis process. A safe and secure storage is provided and maintained by Linköping University. The collecting and processing of personal data from study participants are limited to data necessary to fulfil the objectives of the study. Transcriptions of audio interviews is checked by another person besides the transcriber. In all conversions, maintaining the original information content is ensured. Ethical committee statement and intellectual property rights regulates the storing and opening of research data.

## PATIENT AND PUBLIC INVOLVEMENT
The project is based at Barnafrid—the Swedish National Centre on Violence Against Children in collaboration with Save the Children Sweden and the Refugee Medical Centre. We have particularly chosen to include both public health and non-governmental sectors that can provide us with valuable up-to-date knowledge on the migrants in Sweden as well as to facilitate the dissemination of research findings and translation of the findings into coherent public health policies. No migrant children or young adults are directly involved in the study design, recruitment or conduct of the research. However, the results will be made available to participants, their caregivers, the funders, the professionals working with migrants and researchers.

## ETHICS AND DISSEMINATION
### Ethical aspects
Ethical approval has been granted by the Regional Ethics Board of Linköping (2018/292-31 and 2018/504-32) and the National Ethics Board (2019–05473, 2020–00949 and 2021-03001). Informed consent will be obtained from all study participants and legal guardians if the participant in younger than 15 years. Migrant children are considered a vulnerable group due to an increased risk of having been exposed to violence and it is sometimes suggested that asking about these experiences may cause distress. However, our previous experiences have indicated that

the risk for distress is small. Nevertheless, a plan has been established to take care of participants who need further support.

Participation is based on voluntary informed written consent. The participants, and their legal guardians if the participant is a minor, are informed about the purpose of the study, how the data are protected and stored and most importantly that the interview will have no bearing on their asylum process. They are also informed that they, at any time, can withdraw their consent and exit the study. Incentives valued at approximately €15 or €30 are given to child and adult participants, respectively.

To ensure that the project follows the stated ethical standards, weekly monitoring takes place and ethical issues are documented in a study log.

## Knowledge dissemination and utilization plan
Information about the ongoing study and obtained results are communicated via a homepage (www.barnafrid.se/denlangaresan), lectures, podcasts, interviews and press releases as well as webinars in addition to social media (twitter@denlangaresan).

Community policy-makers and stakeholders will be targeted separately using outreach events to implement new knowledge in current practices and to initiate improvements based on the results. For example, knowledge obtained in this project will be helpful to improve the existing screen of mental health problems and exposure to violence among migrants. Also, an intersectoral process development guide might be constructed to facilitate management and coordination of actions for prevention and intervention of mental health problems.

**Author affiliations**
[1]Barnafrid and Department of Biomedical and Clinical Sciences, Linköping University, Linköping, Sweden
[2]MRC Unit for Lifelong Health and Ageing, University College London, London, UK
[3]Department of Global Public Health, Karolinska Institute, Stockholm, Sweden
[4]Department of Social Sciences, Ersta Sköndal Bräcke University College, Stockholm, Sweden
[5]Department of Child and Adolescent Psychiatry and Department of Biomedical and Clinical Sciences, Linköping University, Linköping, Sweden

**Acknowledgements** Professor Gisela Priebe is acknowledged for contributing to the initial discussions on the study design.

**Contributors** EM, ARK and LK conceived the idea for the study protocol article and drafted the manuscript. EM, ARK, FF, LJ and LK designed the study, critically revised the work and approved the final submitted version.

**Funding** This study is funded by FORTE—the Swedish Research Council for Health, Working Life and Welfare (grant ID: 2019–01660) and was supported by a grant from the Queen Silvia's foundation.

**Competing interests** None declared.

**Patient and public involvement** Patients and/or the public were involved in the design, or conduct, or reporting, or dissemination plans of this research. Refer to the Methods section for further details.

**Patient consent for publication** Consent obtained directly from patient(s)

**Provenance and peer review** Not commissioned; externally peer reviewed.

**ORCID iDs**
Erica Mattelin http://orcid.org/0000-0002-0796-3921
Laura Korhonen http://orcid.org/0000-0002-1837-5930

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
