## [Reviewer comments · BMJ Open]

ARTICLE DETAILS

TITLE (PROVISIONAL)	"The long journey to shelter" - study protocol: a prospective longitudinal analysis of mental health and its determinants, exposure to violence and subjective experiences of the migration process among adolescent and young adult migrants in Sweden
AUTHORS	Mattelin, Erica; Khanolkar, Amal R; Fröberg, Frida; Jonsson, Linda; Korhonen, Laura

VERSION 1 – REVIEW

REVIEWER	Vallejo, Macarena University of Malaga, SOCIAL PSYCHOLOGY, SOCIAL WORK, SOCIAL ANTHROPOLOGY, AND EAST ASIAN STUDIES
REVIEW RETURNED	26-Sep-2020

GENERAL COMMENTS	This is a relevant study that can have a great applicability and be useful to increase the quality of life and well-being of minor migrants residing in Sweden. Furthermore, that this is a longitudinal study and protective factors for mental health, such as resilience, are included are very positive aspects. Below are some aspects to improve the study hoping that they can be helpful. 1. Although the study includes all participants under the category of migrants, legally the situations of immigration, refuge and forced migration are different. I think that these concepts should be defined in the Introduction for greater clarity in the text, and they should not be treated as synonyms. Also, it would also be necessary give some hint of justification about why the study includes all participants in the same category.2. The Introduction should include some references and studies that point to a relationship between social networks and social support with better integration in the host country and a higher quality of life.3. Although in the section on gender perspective it is indicated that the results will be analyzed under a gender perspective, in the Introduction section there is no reference to the possible differences between boys and girls migrants. There is scientific evidence in the literature that, especially in refugee situations, women and girls suffer more violence, particularly sexual violence, than men and boys. In my view, this aspect should be noted in the Introduction and included in the treatment of the data, since greater exposure to violence and traumatic experiences of this nature can lead to worse mental health.
---

	4. Although the study objectives are defined, the hypotheses are not clear since only a general hypothesis is included. A relationship should be established between the factors studied and physical and mental health and social integration in the country. Additionally, some hypothesis could be established regarding the possible differences between the groups (if they are expected) and over time. 5. In the section on participants it would be desirable to indicate the main nationalities of the subjects who have participated in the study. In addition, a separate group corresponding to the 30 subjects should appear in Figure 1, not included in Group 1, since the time taken to apply for asylum is different in each case. 6. There are some errors in the Interviews section that should be corrected. In addition, it is indicated that the arrival of immigrants is by flight, which may not be correct in all cases. 7. When detailing the instruments, no reference is made to how social networks and social support will be measured. 8. It is not clear throughout the article if the process is being carried out at this time or if it will be carried out in the future (in which case the verb tenses are not appropriate since past and futures tenses are mixed). 9. It should be included what results are expected in broad terms from this study and this action protocol and also some future line of research.
--	--

REVIEWER	Fang, Jianqun Ningxia University
REVIEW RETURNED	18-May-2021

GENERAL COMMENTS	This protocol provided a detailed project to explore the adolescent and young adult migrants in Sweden using multiple methods. The aim of this protocol is to understand how different factors impact their mental health and integration into the Swedish society. I have the following questions about this protocol:  1. During the period, what is the specific situation of immigration in Sweden? How did the authors ensure that the participants included in the study were adequately representative? 2. This protocol showed that there were only 160 participants left in this study after one year. With such a high rate of lost to follow-up, how can you prove that the results of this study still be representative and of social value? 3. Do you think there are confounding factors for immigrants from different countries due to cultural background, religious belief, economic level, political system and other issues? In the process of statistical analysis, how to determine the relevant confounding factors and whether to control the interference caused by those factors? 4. Page 6, what is the basis for this sample size estimate of 30 subjects in qualitative interviews? How is its representativeness ensured?
---

VERSION 1 – AUTHOR RESPONSE

Reviewer: 1
Dr. Macarena Vallejo, University of Malaga
Comments to the Author:

This is a relevant study that can have a great applicability and be useful to increase the quality of life and well-being of minor migrants residing in Sweden. Furthermore, that this is a longitudinal study and protective factors for mental health, such as resilience, are included are very positive aspects.

Below are some aspects to improve the study hoping that they can be helpful.

1. Although the study includes all participants under the category of migrants, legally the situations of immigration, refuge and forced migration are different. I think that these concepts should be defined in the Introduction for greater clarity in the text, and they should not be treated as synonyms. Also, it would also be necessary give some hint of justification about why the study includes all participants in the same category.

Response:

We wish to thank the reviewer for this comment. We have clearly defined our population in the section on participants (page 5-6). The reason for including asylum seekers, quota refugees and family reunification migrants is that they are expected to have similar experiences of trauma, and migration journey. If we were to create smaller groups e.g., to only include quota refugees, we would not reach statistical power due to the small number of asylum-seekers that arrive to Sweden today.

2. The Introduction should include some references and studies that point to a relationship between social networks and social support with better integration in the host country and a higher quality of life.

Response:

Thank you for pointing out this shortcoming. We have added new references at the page 5.

3. Although in the section on gender perspective it is indicated that the results will be analyzed under a gender perspective, in the Introduction section there is no reference to the possible differences between boys and girls migrants. There is scientific evidence in the literature that, especially in refugee situations, women and girls suffer more violence, particularly sexual violence, than men and boys. In my view, this aspect should be noted in the Introduction and included in the treatment of the data, since greater exposure to violence and traumatic experiences of this nature can lead to worse mental health.

Response:

We thank you for this valuable comment. We do think this is a very important topic and have inserted a sentence about this in the introduction. Please, see page 5. As noted, the gender aspect will be included in the treatment of the data.

4. Although the study objectives are defined, the hypotheses are not clear since only a general hypothesis is included. A relationship should be established between the factors studied and physical and mental health and social integration in the country. Additionally, some hypothesis could be established regarding the possible differences between the groups (if they are expected) and over time.

Response:

We thank the reviewer for this valuable comment. We have revised the section Objectives according to the above input.

5. In the section on participants it would be desirable to indicate the main nationalities of the subjects who have participated in the study. In addition, a separate group corresponding to the 30 subjects

should appear in Figure 1, not included in Group 1, since the time taken to apply for asylum is different in each case.

Response:

Thank you for this comment. The nationalities of the participating children will be reported. We still do not know this detail as data collection is currently ongoing. We have also adjusted Figure 1 accordingly.

6. There are some errors in the Interviews section that should be corrected. In addition, it is indicated that the arrival of immigrants is by flight, which may not be correct in all cases.

Response:

Thank you. We have re-checked the ms and corrected errors that we have identified. 'flight' is not referring to travel by airplane but to the process of fleeing one's country of origin. We have changed the wording to enhance clarity. Please, see page 7

7. When detailing the instruments, no reference is made to how social networks and social support will be measured.

Response:

Social networks and social support will be measured by questions referred to in Table 1. A line illuminating this has been inserted in the interview-section (page 7).

8. It is not clear throughout the article if the process is being carried out at this time or if it will be carried out in the future (in which case the verb tenses are not appropriate since past and futures tenses are mixed).

Response:

We wish to thank the reviewer for pinpointing this inconsistency. The process is being carried out right now and will continue until 2022. We have checked the verb tenses and changed that accordingly.

9. It should be included what results are expected in broad terms from this study and this action protocol and some future line of research.

Response:

We thank the reviewer for this suggestion and have added this information on the page 5.

Reviewer: 2

Dr. Jianqun Fang, Ningxia University

Comments to the Author:

This protocol provided a detailed project to explore the adolescent and young adult migrants in Sweden using multiple methods. The aim of this protocol is to understand how different factors impact their mental health and integration into the Swedish society.

I have the following questions about this protocol:

1. During the period, what is the specific situation of immigration in Sweden? How did the authors ensure that the participants included in the study were adequately representative?

Response:

We wish to thank the reviewer for this comment. In 2020, approximately 13 000 asylum seekers, including 3566 children, came to Sweden. Most of the asylum seekers originate in Syria, Afghanistan,

Iraq, and Eritrea. We monitor continuously the national statistics on asylums seekers provided by the Migration Agency (<https://www.migrationsverket.se/English/About-the-Migration-Agency/Statistics.html>) and have targeted recruitments of different groups in order to ensure the best possible representativity. This information is now also stated on page 6.

2. This protocol showed that there were only 160 participants left in this study after one year. With such a high rate of lost to follow-up, how can you prove that the results of this study still be representative and of social value?

Response:

We thank the reviewer for pointing out the need to clarify: "A subset (n=160) will be interviewed again after one year". This sentence refers to adolescent subsample (n=160) that will be interviewed at baseline and one year afterward. We have re-phrased this sentence as follows: "A subsample of adolescents (n=160) will be re-interviewed after one year.". We have also clarified within the section Design and participants. Please, see page 6.

3. Do you think there are confounding factors for immigrants from different countries due to cultural background, religious belief, economic level, political system and other issues? In the process of statistical analysis, how to determine the relevant confounding factors and whether to control the interference caused by those factors?

Response:

We appreciate that the reviewer points out this important issue. Previous studies have indicated that cultural, economy-related, and political factors play a role in others in the decision to flee and integrate into the new society. We have chosen not to specifically address these issues because translator assisted interviews set a limit to the amount of data that can be collected, and strict priority has been given to most well-known confounders (e.g., gender, sex, and SES) for mental health and exposure to violence, the primary interest of this study. In addition, our experience from already conducted interviews is that many children even have difficulties to report their ethnicity. Keeping with this, we expect that the collection of high-quality data on economy-related factors and political systems might be very difficult.

4. Page 6, what is the basis for this sample size estimate of 30 subjects in qualitative interviews? How is its representativeness ensured?

Response:

We thank the reviewer for this comment. We are finished with this data collection and have included younger and older adolescents, both genders, those migrating with their families, and unaccompanied minors. Altogether, participants represent 12 different countries and four regions. A sentence on how to sample representativity is ensured has been added on page 6.